# Bacterial 2′-Deoxyguanosine Riboswitch Classes as Potential Targets for Antibiotics: A Structure and Dynamics Study

**DOI:** 10.3390/ijms23041925

**Published:** 2022-02-09

**Authors:** Deborah Antunes, Lucianna H. S. Santos, Ernesto Raul Caffarena, Ana Carolina Ramos Guimarães

**Affiliations:** 1Laboratório de Genômica Funcional e Bioinformática, Instituto Oswaldo Cruz, Fundação Oswaldo Cruz, Rio de Janeiro 21040-900, Brazil; ana.guimaraes@fiocruz.br; 2Laboratório de Modelagem Molecular e Planejamento de Fármacos, Departamento de Bioquímica e Imunologia, Instituto de Ciências Biológicas, Universidade Federal de Minas Gerais, Belo Horizonte 31270-901, Brazil; luciannahss@rpd.ufmg; 3Grupo de Biofísica Computacional e Modelagem Molecular, Programa de Computação Científica, Fiocruz, Rio de Janeiro 21040-360, Brazil; ernesto.caffarena@fiocruz.br

**Keywords:** 2′-deoxyguanosine riboswitch, purine riboswitch, molecular dynamics simulation, network analysis, ligand binding mechanism

## Abstract

The spread of antibiotic-resistant bacteria represents a substantial health threat. Current antibiotics act on a few metabolic pathways, facilitating resistance. Consequently, novel regulatory inhibition mechanisms are necessary. Riboswitches represent promising targets for antibacterial drugs. Purine riboswitches are interesting, since they play essential roles in the genetic regulation of bacterial metabolism. Among these, class I (2′-dG-I) and class II (2′-dG-II) are two different 2′-deoxyguanosine (2′-dG) riboswitches involved in the control of deoxyguanosine metabolism. However, high affinity for nucleosides involves local or distal modifications around the ligand-binding pocket, depending on the class. Therefore, it is crucial to understand these riboswitches’ recognition mechanisms as antibiotic targets. In this work, we used a combination of computational biophysics approaches to investigate the structure, dynamics, and energy landscape of both 2′-dG classes bound to the nucleoside ligands, 2′-deoxyguanosine, and riboguanosine. Our results suggest that the stability and increased interactions in the three-way junction of 2′-dG riboswitches were associated with a higher nucleoside ligand affinity. Also, structural changes in the 2′-dG-II aptamers enable enhanced intramolecular communication. Overall, the 2′-dG-II riboswitch might be a promising drug design target due to its ability to recognize both cognate and noncognate ligands.

## 1. Introduction

The spread of antibiotic-resistant bacteria represents a substantial health threat contributing to morbidity and mortality worldwide [1]. In 2016, the World Health Organization (WHO) requested the Member States to create a priority list of antibiotic-resistant bacteria to support the research and development of effective drugs [2]. Current antibiotics act on a few metabolic pathways, facilitating the emergence of antibiotic-resistant bacteria. Consequently, developing new regulatory inhibition mechanisms that act differently from the current ones becomes an immediate necessity. In this scenario, riboswitches represent novel and promising targets for developing useful antibacterial drugs. These non-decoding RNA structures regulate up to 4% of all bacterial genes, many of which are essential for their survival [3].

Riboswitches are RNA sensors that affect post-transcriptional processes by binding to small molecules, such as vitamins, amino acids, and nucleotides [4,5]. Their structure entails two domains, the sensory and the regulatory domains. The first is the aptamer, which acts as a receptor for specific metabolites, and whose sequence and structure are highly conserved. Adjacent to the aptamer, is the expression platform, which is responsible for transducing small molecule binding to genetic regulatory signals. Transduction is achieved by a structural rearrangement, resulting in an immediate RNA conformational change that alters mRNA translation [6].

Because of the structural conservation of the riboswitch aptamer, we can speculate that a drug similar to the metabolite could bind to the riboswitch, inactivating the expression of a vital gene [7]. Since some riboswitches exist in multiple genes or control entire operons, collective repression of these genes in a given metabolic pathway can be deleterious. Depending on their natural distribution, drugs that target riboswitches can become potential broad-spectrum antibiotics (such as drugs targeting thiamine pyrophosphate (TPP) or flavin mononucleotide (FMN) riboswitches) or as more selective drugs (such as purine riboswitches) [8].

The purine riboswitch family, which regulates aspects of purine biosynthesis and transport in bacteria, recognizes and binds to different ligands like guanine [9], adenine [10], or 2′-deoxyguanosine [11,12]. The aptamer domain folding and purine recognition pattern are highly similar within this family of riboswitches. Crystallographic structures revealed that all purine riboswitches maintain their architecture regardless of the ligand they recognize [13]. Furthermore, the consensus sequence of these identified riboswitches deviates from one another in up to three places in the regions directly involved in the ligand recognition [14]. Since they recognize multiple ligands, the purine riboswitch family can be used as a model for understanding recognition mechanisms.

Among the purine riboswitch classes, there are two different 2′-deoxyguanosine riboswitches, known as 2′-dG-I and 2′-dG-II. The 2′-dG-I riboswitch was identified in *Mesoplasma florum* and is located in the 5′ region of the gene encoding the β subunit of the ribonucleotide reductase protein [15]. This enzyme converts phosphorylated ribonucleotides into their corresponding phosphorylated deoxyribonucleotides, which make up the building blocks of DNA [16]. The 2′-dG-II riboswitches have been supposed to detect dG as the natural ligand [14]. Genes downstream from 2′-dG-II riboswitches with assigned functions are predicted to encode a signal receiver domain, phospholipase D, ComEC, and endonuclease I. The last two work on DNA substrates.

Interestingly, the bacterium *M. florum*, which carries many 2′-dG-I riboswitches, is classified in the phylum Tenericutes, whereas 2′-dG-II riboswitches are found in the phylum Bacteroidetes. Thus, these 2′-dG riboswitch classes are present in highly diverged organisms and can be regarded as sequence-discrepant entities. These features show that two different evolutionary events might have happened when the classes evolved from guanine riboswitches [14].

In this context, understanding the mechanism of ligand recognition is crucial for proposing a riboswitch as an antibiotic target. To comprehend how modification of similar RNA sequences might alter selectivity to bind alternative small molecules, we used a combination of computational biophysics approaches to investigate the structure, dynamics, and the energy landscape of 2′-dG riboswitch classes bound to 2′-deoxyguanosine (dG) and noncognate riboguanosine (rG). Our results suggest that the stability and increased interactions in the three-way junction of 2′-dG riboswitches are related to a higher affinity for nucleoside ligands. We also highlighted that critical structural differences in the 2′-dG-II aptamers enable enhanced intramolecular communication. This riboswitch class appeared to be a promising target for drug design since it presents high affinity and specificity for both cognate and noncognate ligands.

## 2. Results

### 2.1. RNA Content Analysis

Nucleotide sequence alignment between 2′-dG-I and 2′-dG-II riboswitches revealed 36.7% identical residues (Figure 1A), along with conserved structures. Like all members of the purine riboswitch family, the aptamer comprises a three-way junction formed by regulatory helix P1 and P2-L2 and P3-L3 hairpins connected by three junctions and stabilized by loop–loop interactions (Figure 1B).

The ligand-binding pocket is located at the center of the three-way junction structure (Figure 1C). The 2′-dG-I contains a first shell of interacting nucleotides C31, C58, and C80, located approximately <3 Å, similar to equivalent nucleotides U22, C51, and C74 in the 2′-dG-II riboswitch. Cytosine at position 58/51 of the 2′-dG riboswitches classes is critical for establishing selectivity for nucleoside against nucleobasis across the guanine/adenine classes, which has uracil in the same position [11,12,14].

The two classes of 2′-dG binders present other significant differences around the ligand-binding pocket that distinguish from one another. For example, in the 2′-dG-II riboswitch, the junction-proximal base pair in P1 consists of a G–C–pair, whereas, for the 2′-dG-I and other purine riboswitch classes, a nearly invariant A–U pair forms. Another distinction between the 2′-dG-II and other classes in the purine family is the insertion of two nucleotides (U75 and C76) within J3/1 (Figure 1B).

Differences beyond the ligand-binding pocket may also be significant for affinity and selectivity of the 2′-dG classes for a particular ligand. For example, an (A-U) Watson–Crick pair closing P2 in the 2′-dG-II class replaces the C-G pair at the same position in 2′-dG-I. However, the main architectural difference between the two RNA aptamers consists of the smaller sequence content of the L3 in 2′-dG-I, compared to 2′-dG-II, which changed the conformation and interface of the kissing loops (4 nucleotides) compared with those in the purine riboswitches (7 nucleotides) (Figure 1B). Nevertheless, both dG riboswitches preserved two tertiary Watson-Crick G-C bases between L2 and L3.

Superposition between the 2′-dG-I and 2′-dG-II crystal structures, using the RNA-align algorithm for comparing 3D structures of RNA molecules, resulted in a root-mean-square deviation (RMSD) of 3.43 Å and a TM-score of 0.43. It is worth mentioning that TM-score assesses the structural similarity on a scale from 0 to 1, where 1 indicates a perfect match and a score ≥ 0.45 corresponds to the structural similarity of the RNA pairs in the same Rfam family. Thus, this TM-score value below the threshold can be mainly associated with the differences in the P3-L3 stem-loop (Figure 1C).

### 2.2. Influence of Ligands on Dynamics of 2′dG Riboswitches

RMSD and root-mean-square fluctuation (RMSF) values were computed to evaluate the influence of the cognate dG and the noncognate rG (Figure 1D) on 2′-dG riboswitch stability from molecular dynamics simulations. We regarded only the heavy atoms for calculations over the trajectory production stage, taking as references the crystallographic structures (Table 1 and Figure 2). The RMSD values of the whole aptamer of the two classes of riboswitches were equivalent to a meaningless difference below 0.5 Å (Table 1). The overall fluctuation pattern was similar for all systems, with the highest peaks in the P1, L2, J2/1, and L3 regions (Figure 2). However, the inspection of some particular regions of the RNA aptamer evidenced slight differences in their dynamical behavior.

The presence of ligands impacted 2′-dG-I aptamer fluctuation. Nucleotide fluctuation was higher in the dG-bound system, especially in the P3-L3-J3/1 region (Figure 2A,B). Unlike the class-I riboswitches, those belonging to the 2′-dG-II class remained unaltered, regardless of being bound to dG or rG. The 2′-OH group of rG contributed to enhancing the flexibility of J2/3 and L3 and slightly reduced flexibility of P1 in 2′-dG-II aptamers (Figure 2C,D).

When analyzing ligand stability in the 2′-dG-II riboswitches, the cognate dG remained steadier than the noncognate rG, with 1.6 ± 1.0 Å and 2.2 ± 1.1 Å, respectively. The opposite was observed for those belonging to class I, with an RMSD of 3.0 ± 1.0 Å and 2.1 ± 0.5 Å for dG and rG, respectively. As already reported, the 2′-OH group of the ribose ligand fits the binding pocket by assuming a favorable conformation C3′-*endo* rather than the C2′-*endo* conformation, typically observed in deoxyribonucleosides [11].

Thus, we also evaluated the sugar pucker of the ligands along the trajectories (Figure 3). The higher fluctuation of dG-bound in the 2′-dG-I riboswitch affected sugar puckering conformations. Although mostly adopting the C2′-endo conformation, it constantly interchanged with the opposite C2′-exo and C3′-endo conformations (Figure 3A). In contrast, dG bound on the 2′-dG-II riboswitch remained in the C2′-endo conformation (Figure 3C). Ligands associated with the rG-bound aptamers behaved similarly, keeping the C3′-endo conformation (Figure 3B,D) along with an RMSD of ~2 Å of rG.

### 2.3. Analysis of Binding Free Energy

Binding free energy calculation is essential to clarify ligand association and recognition phenomena. In addition, it is usually applied to quantify the interaction intensity between receptor and ligand, thus ranking ligands based on individual affinities [17]. To understand how structural modifications in the 2′-dG riboswitches might alter the selectivity for the dG and rG, we have calculated the absolute binding free energy (ΔG_bind_) using the MM/GBSA method [18].

Several assessments upon different aspects of the 2′-dG riboswitches were elucidated through our simulations. Overall, evaluations of all systems resulted in negative ΔG_bind_ values, indicating spontaneity in the recognition process (Table 2). Of both 2′-dG riboswitches, the class II predicted binding affinity reproduced the experimental data satisfactorily. The prediction for the rG binding was almost the same as the reported experimental value, approximately −11 kcal/mol [12], while calculations for dG were not so accurate, and a deviation of almost 6 kcal/mol was observed when compared with the experimental one (−11.6 kcal/mol). When analyzing the ΔG_bind_ decomposition into individual energies in the MM/GBSA method, we noted that the 2′-dG-II riboswitch showed a higher affinity for dG than rG, mainly due to its van der Waals contribution (ΔE_vdw_).

On the other hand, the ΔG_bind_ prediction for the 2′-dG-I riboswitches did not agree in absolute value or in the ranking with the previous experimental data. The prediction for dG was −6.4 kcal/mol versus the experimental value of −10.9 kcal/mol (|Error| = 4.5), while the noncognate rG predicted a ΔG_bind_ of −16.0 kcal/mol versus −8.6 kcal/mol (|Error| = 7.4) [11]. The electrostatic energy term (∆G_ele+egb_) contributed mainly to lowering the dG binding affinity with an associated value of 16.0 kcal/mol. This same contribution was 8.9 kcal/mol for rG, a value shared by both ligands bound to the 2′-dG-II riboswitches (~10 kcal/mol).

In agreement with experimental data, dG can bind to 2′-dG-II riboswitch with higher intensity than to 2′-dG-I riboswitch [11,12]. In our simulations, the entropic contributions (−T∆S) resulted in comparable values for both systems (23 kcal/mol), so the divergences in ΔG_bind_ observed among the classes were due to the enthalpic term, and consequently, the binding process was enthalpically driven. However, a slight difference in the −T∆S term for rG bound to both classes of riboswitches could be observed, being 21 kcal/mol for rG bound to 2′-dG-I compared to ~23 kcal/mol in all other complexes.

### 2.4. The Purine Riboswitch Family Share Identical Critical Nucleotides to Ligand Binding

The MM/GBSA method also supplied the decomposition of the binding energy for each nucleotide. From these values, we examine how the binding site nucleotides could contribute to the total binding energy. In agreement with previous in silico surveys on guanine [19,20] and adenine [21] riboswitches, we detected that the six equivalent nucleotides [A30(G21), C31(U22), C58(C51), G59(A52), C80(C78), U81(C79)—in parentheses are the corresponding nucleotides in class II] present favorable average energy values less than −1.0 kcal/mol in most complexes. Furthermore, out of these nucleotides, two of them [C58(C51) and C80(C78)] contributed mainly electrostatically, while for the others, the van der Waals contribution was dominant. (Figure 4).

The cytosine located at position 58/51, which is critical for establishing selectivity for nucleoside, contributed most to the affinity of both dG and rG ligands. This nucleotide from the 2′-dG-I riboswitch, in complex with dG, presented the lowest energy value (−4.77 kcal/mol). Simulations showed that cytosine could hydrogen-bond the ligands extensively. Notably, C58 and dG formed up to five hydrogen bonds (H-bonds) during the last 50 ns with an occupancy higher than 21% of this simulation time (Figure 5A). The main interactions happened between C58@O2 and LIG@N2, and LIG@N3 with 98% and 83% occupancy, respectively. The same riboswitch bound to rG showed a slight increase in energy (−4.28 kcal/mol) and fewer H-bonds (up to 3), with the pairs C58@N3–LIG@N2 and C58@N4–LIG@N3 with 99% occupancy. The presence of the 2′-OH group in rG induced the formation of an extra H-bond with C58@N4 with an occupancy of 48.5%. In the 2′-dG-II riboswitches, C51 displayed a similar energy value of −3.5 kcal/mol for both ligands. The rG interactions were equivalent in class I riboswitches, considering atom pairs and occupancy values. On the other hand, dG favored the pairs C51@O2-LIG@N2, as in class I, and C51@N3–LIG@N2.

As already mentioned, cytosine 80/78 also plays a crucial electrostatic role through H-bonds formation. Decomposition values below −4 kcal/mol in rG-bound 2′-dG riboswitches were related to an increase in the H-bonds number compared to dG-bound systems (approximately −1.2 kcal/mol) (Figure 4 and Figure 5). The first nucleotide of J1/2 [C31(U22)] exhibited a more negative energy contribution when bound to rG. Despite being influenced mainly by the van der Waals energy term, approximately −3.2 kcal/mol in all systems, the electrostatic energy achieved negative contributions for rG (−0.71 kcal/mol and −2.04 kcal/mol for classes I and II, respectively) and positive for dG (4.74 kcal/mol and 0.60 kcal/mol for classes I and II, respectively). This may be connected to the H-bond formation that enhances occupancy between C31(U22)@O2′ and LIG@O6 in the rG-bound riboswitches (Figure 5).

The 2′-dG-II riboswitch bound to the dG has two energetically relevant nucleotides, G21 and C50 (Figure 4). The formation of two singular H-bonds can justify the lower energy value of −6.71 kcal/mol of the G21 with an occupancy higher than 94% for the G21@N2–LIG@O4′ and G21@N3–LIG@O5′ pairs (Figure 5). Likewise, C50, with energy of −2.63 kcal/mol, formed an H-bond between C50@N4–LIG@O4′ with an occupancy of 83%.

### 2.5. Cognate and Noncognate Ligands Alter the Dynamical Secondary Structure

The structure and dynamics of nucleic acids are affected by two types of noncovalent interactions: H-bond and aromatic stacking [22]. Thus, we evaluated the riboswitches’ secondary structure evolution along the trajectories to identify variability in the annotation within a set of structures (Figure 6).

Among the riboswitch classes, the most notable difference appeared by analyzing the interactions between the L2 and L3 loops. Both classes share two tertiary Watson-Crick G-C bases between L2 and L3, preserved over time with an occupancy higher than 92% of the simulation time. The 2′-dG-I riboswitches contain a shorter L3 loop, affecting the number of interactions between loops (3 base pairs and 3 stackings) compared to class II (8 base pairs and 2 stackings). Neither dG or rG altered the interactions between loops in class I. The only difference observed was the interchangeability between the cis and trans conformations of the Sugar/Hoogsteeen pairing A40–A71 in the rG-bound system (Figure 6A,B). The 2′dG-II riboswitches’ systems showed the same interaction patterns, but the occupancy of interactions between loops was weakened by approximately 9% in the rG-bound system (Figure 6C,D).

The three-way junction structure was another region displaying significant differences between the riboswitch classes. The U75 from the J3/1 region, exclusive to class II, favored the Hoogsteeen/Watson-Crick interaction with A24 from J1/2. This adenine also stacked C54 (P3) and A77 (J3/1), strengthening interactions between the junctions that form the binding pocket (Figure 6C,D). In class I, G33, instead of A24, formed a pairing with U52 of P2, not contributing to interactions between the junctions (Figure 6A,B).

As the ligands are nucleosides, it was also possible to evaluate their interactions with RNA aptamers. Cytosine at position 58/51, which is critical for establishing selectivity for nucleoside, interacted through a Watson-Crick/Sugar bond with the ligand with an occupancy greater than 97% in all systems. Stackings between the two ligands and G59(A52) and U81(C79) were also preserved. The stacking between ligands and U22 only occurred in the 2′-dG-II riboswitches.

The significant discrepancy between the aptamer and nucleoside ribose and deoxyribose is related to cytosine 80/78, which formed the Watson-Crick base pair with rG. This result agrees with decomposition values below −4 kcal/mol in rG-bound 2′-dG riboswitches (Figure 4). We found a Sugar/Watson-Crick interaction between A52 and C78 with 70% occupancy in the 2′-dG-II riboswitch bound to dG, preventing C78 and dG pairing (Figure 6C). A similar scenario could be seen in the 2′-dG-I riboswitch, keeping both pairings between C80 and dG and G59, with 60% and 27% occupancy, respectively (Figure 6A). In rG-bound systems, there was no pairing between C80/78 and G59(A52) (Figure 6B,D).

### 2.6. Anticorrelated Motions Are Changed by Cognate and Noncognate Ligands

We calculated the dynamic cross-correlation matrices (DCCM) between pairs of nucleotides to investigate how dG and rG binding affects the dynamic couplings in both 2′-dG riboswitches. Overall, a similar cross-correlation pattern was observed for all systems. As expected, the strongest positive correlations (≥0.6) occur between the three helical strands P1, P2, and P3 (Figure 7). The 2′-dG-I riboswitch bound to dG displayed the greatest extent and magnitude of anticorrelations, with connections occurring mostly between the P1 and J1/2, J2/3, and P3 regions (Figure 7A), mainly due to the higher flexibility class-I riboswitches than the others (Table 1). Conversely, the rG-bound in class II increased anticorrelations intensity, especially between J3/1 and the second strand of P1, compared with the cognate (Figure 7D).

The two additional nucleotides in J3/1, unique to the 2′-dG-II riboswitch, enhanced the magnitude of the positive correlations with J1/2 over class I (Figure 7C,D). The 2′-dG-II riboswitch, when bound to rG, decreases the positive correlations between L2 andL3 regarding the cognate ligand. This result aligns with dynamic secondary structure analysis, revealing decreased occupancy of L2–L3 interactions (Figure 6). In the 2′-dG-I riboswitch, the nucleoside ligands did not modify the correlations between the loops (Figure 7A,B). All ligands showed a strong positive correlation with the junction-proximal base pair in P1, J1/2, J2/3, and J3/1. These regions comprise the binding site residues that contributed the most to total binding energy (Figure 2).

### 2.7. Communication Pathways between P1–P3

To get a deeper understanding of how the ligands and distinct nucleotides between classes of 2′-dG riboswitches influence global aptamer communication, we computed the 1000 shortest paths between A30(G21) and C39(A31) of junction-proximal base pair in P1, and the base pair closing P2, respectively. For this, we performed a correlation network analysis by constructing weighted graphs in which a single node represents each nucleotide, and the weight of the connection between pairs of nodes was proportional to their respective previously calculated correlation coefficients (Figure 7). We also calculated the normalized node degeneracy metric using the shortest paths, which revealed the percentage of paths accessing each node. (Figure 8).

The distribution of node degeneracies differed between 2′-dG riboswitches classes, showing that nucleotide variations in the aptamer domain alter intramolecular communication. For 2′-dG-II riboswitches, the found distributions were very similar, showing that the type of ligand did not alter the communication between nucleotides (Figure 8C,D). In contrast, the dG-bound 2′-dG-I riboswitch increased the degeneration of the P3 and J3/1 regions compared with the rG-bound system (Figure 8A,B). To quantify this analysis, we calculated the square inner product (SIP) [23] to compare the overall similarity of the node degeneracy profiles between dG-bound and rG-bound systems. Accordingly, high SIP values are associated with a remarkably similar intramolecular communication between ligands. We obtained a higher SIP for 2′-dG-II riboswitches (0.97) than for 2′-dG-I riboswitches (0.63), reinforcing that the 2′-OH group of rG did not change the modulation of intramolecular communication in class II.

In both 2′-dG-II riboswitches, the critical nucleotide U22 was essential for communication between P1 and P2, with approximately 0.97 degeneracy. Therefore, the preferred communication pathway involved the aptamer’s P1–J1/2–P2 regions (Figure 8C,D). In class I, the equivalent nucleotide (C31) resulted in a degeneration of 0.64 and 0.18 for rG and dG, respectively. Here, the key nucleotide C80, from the J1/3 region, presented a degeneration of 0.72 and 0.33 for dG and rG, respectively. Moreover, the preferred communication path of the dG-bound system involved the P1–J3/1–P3–L2–P2 regions (Figure 8A,B).

Additionally, we calculated the number of nodes per path (Figure 9). Indeed, ligand type did not change the number of nodes per path in the 2′-dG-II riboswitches of about 11/12 nucleotides. However, in 2′-dG-I riboswitch systems, we noticed a bifurcation with paths involving about 10 and 19 nucleotides. In the dG-bound system, most pathways involved 20 nucleotides, supporting communication, including more regions of the aptamer (P1–J3/1–P3–L2–P2), while in the rG-bound system, the preference was for pathways involving about 11 nucleotides, like class II systems, and communication through the aptamer’s P1–J1/2–P2 regions.

## 3. Discussion

In this work, we performed 8 μs molecular dynamics simulations on four systems involving the 2′-dG-I and 2′-dG-II riboswitches complexed with two ligands, cognate dG and the noncognate rG, to study the ligands’ recognition mechanism. Although the 2′-dG-I and 2′-dG-II aptamers possess a low sequential identity (36.7%), they share similar core structures to bind to the same ligand.

Cytosine at position 58/51 constitutes a similar fundamental aspect between the classes of 2′-dG riboswitches. As already mentioned, C58/51 is critical for establishing selectivity for nucleoside against nucleobasis across the guanine riboswitch, which has uracil at the same position [11,12,14]. Mutation of C58/51 to uracil causes significant loss of ligand binding affinity of 83-fold [11] and 18-fold [12] for 2′-dG-I and 2′-dG-II riboswitches, respectively. Additionally, the U51C single-point mutation in the guanine riboswitch suffices to switch the ligand preference from guanine to dG [24]. Furthermore, this cytosine contributed mainly to the affinity of both dG and rG ligands with energy less than −4.2 kcal/mol for 2′-dG-I and −3.5 kcal/mol for 2′-dG-II according to the MM/GBSA analysis (Figure 4). C58 of 2′-dG-I connects with the ligand’s sugar edge [11,24] similarly to the equivalent base, C51, in 2′-dG-II riboswitch [12]. This result agrees with the dynamic secondary structure outcomes in which the interaction between C58/51 and both ligands facilitates Watson-Crick/Sugar pairing, with an occupancy greater than 97% in all systems (Figure 6).

In the purine riboswitch family, the discrimination between purines is caused by only one mutation, where cytosine 74 in the guanine riboswitch (equivalent to 80/78 in 2′-dG) corresponds to a uracil 74 in adenine riboswitch, that forms the Watson-Crick pairing with guanine and adenine, respectively [9,25]. In the 2′-dG riboswitches, C80/78 forms a Watson-Crick base pair with the 2′-deoxyguanosine nucleobasis [11,12] and is essential for high-affinity ligand binding; mutating to adenosine results in 1200-fold weaker binding affinity [12]. Although C80/78 is crucial to distinguish between the purine bases and other nucleotides of the pocket, we identified differences concerning the dG and rG ligands. In rG-bound aptamers, the Watson-Crick base pair was maintained and contributed to Gibb’s energy decomposition with values below −4 kcal/mol (Figure 4 and Figure 6). In 2′-dG-II riboswitch dG-bound, C78 changed pairing with dG by Sugar/Watson-Crick interaction with A52, strengthening connections with the three-way junction structure (Figure 6C). In the 2′-dG-I riboswitch, we can observe an intermediate scenario, keeping both pairings between C80 and dG and G59 (Figure 6A).

During the recognition of dG and rG, the 2′-dG-I riboswitch was initially assumed to bind to the rG with weaker affinity because of a steric clash between the RNA and the additional guanosine 2′-OH group [15]. However, after resolving the crystal structure of the 2′-dG-I riboswitches bound to dG and rG, it was possible to observe that the 2′-OH group of rG accommodates itself in the binding pocket by assuming the C3′-*endo* conformation of ribose, instead of the C2′-*endo* conformation of deoxyribose [11]. Our results showed that the sugar puckering of the rG maintained the favorable C3′-endo conformation in both classes of riboswitches, and the dG-bound in the 2′-dG-II riboswitch maintained the C2′-*endo* conformation. However, dG-bound in the 2′-dG-I riboswitch constantly interchanged between the opposite C2′-*exo* and C3′-*endo* conformations (Figure 3). Possibly, this could explain the lower affinity of this system (−6.49 kcal/mol) with the rG-bound in the 2′-dG-I riboswitch (−16.08 kcal/mol).

It was previously reported that the C3′-*endo* conformation of ribose is incompatible with hydrogen bonds from 3′-OH to C56 in class I, and this lack of interaction likely destabilizes the J2/3, decreasing guanosine binding by a factor of 50 [11]. We observed that the 3′-OH group of dG interacts with C56 (U49) with an occupancy of 48% and 98% for classes I and II, respectively. The rG-bound systems also showed interactions with C56 (49) but with other ribose hydroxyl groups (2′-OH and 5′-OH) and with a maximum occupancy of 41% (Figure 5). Pikovskaya et al. have also shown that destabilization of J2/3 affects discrimination against various dG analogs and likely perturb genetic control in the 2′-dG-I riboswitch system [11]. We have identified a correlation between J2/3 stability and binding free energy. In class II, J2/3 displayed greater stability when bound to dG (Table 1), with higher affinity than rG-bound (−17.32 kcal/mol and −11.90 kcal/mol, respectively). Ligands caused the opposite impact on 2′-dG-I aptamers.

Both classes of riboswitches have singularities that allow them to bind to nucleoside ligands. The class I riboswitches required local and distal modifications to have a greater affinity for nucleosides. The 2′-dG-I G33 nucleotide in the three-way junction formed a pairing with the U52 of P2 (Figure 6A,B), not observed in the other purine riboswitches. Another distinction is that the L2—L3 interaction that preserves the Watson-Crick G-C pairs between the loops differs substantially otherwise (Figure 6). The guanine riboswitches with loops akin to 2′-dG-I riboswitch loops showed a reduction in guanine binding by a factor of >50 [11,15]. Also, the 2′-dG-I riboswitch mutant reduced dG binding by a factor of 2.9, where the natural kissing loops were replaced by those from the guanine riboswitch [24].

In contrast, the 2′-dG-II class appears to exploit only local differences around the ligand-binding pocket to achieve a high affinity of nucleosides. The two-nucleotide insertion in J3/1 (U75 and C76) resulted in a moderate loss of affinity for dG (7.1-fold), while a U75A mutant reduced affinity 46-fold [12]. Inserting these two nucleotides into the guanine riboswitch aptamer at the equivalent position does not promote 2′-dG binding [12]. Matyjasik and Batey evidenced that these insertion elements are not essential for promoting dG binding in 2′-dG-II class of RNA, although the A24-U75 interaction helps increase affinity [12]. We corroborate this evidence with the stable Hoogsteeen/Watson-Crick interaction between A24 and U75 (Figure 6C,D).

Additionally, the two extra nucleotides in J3/1 increased the magnitude of the positive correlations, strengthening the connections between the three-way junction (Figure 7). Another essential particularity of the 2′-dG-II aptamer is G21-C79 of the junction-proximal base pair in P1 that has higher affinity for dG. The G21A-C79U mutation decreased the affinity for dG by 16.9 times than for wild-type [12]. These results are in line with the nucleotide’s free energy decomposition, in which the G21 of dG-bound in the 2′-dG-II riboswitch stands out with the lower energy value of −6.71 kcal/mol (Figure 4).

Based on the evidence presented here, we identified the relevance of stability and increased interactions in the three-way junction of 2′-dG riboswitches for higher affinity for nucleoside ligands. DCCM data also supported these outcomes by showing that the high anticorrelated movements between different regions of the aptamer are related to lower affinity for the ligand (Figure 7). For example, 2′-dG-I riboswitch bound to the dG, which showed marked anticorrelated movements, had the lowest affinity (−6.49 kcal/mol). The class II aptamer bound to the dG, which increased the magnitude of the positive correlations, had a higher affinity of −17.32 kcal/mol. The 2′-dG-II riboswitches include an additional set of interactions resulting in significantly higher affinity for their nucleoside binding than their 2′-dG-I counterparts [12]. Here, we have shown through network analysis that critical structural differences in the 2′-dG-II aptamers enable enhanced intramolecular communication (Figure 8 and Figure 9).

In the context of these RNAs for biological applications as drug design, the 2′-dG-II riboswitch appeared to be a promising target since it presents high affinity and specificity for both cognate and noncognate ligands. Furthermore, the discovery of riboswitches paved the way to develop potential antibiotics targeting cellular mechanisms not challenged by current antibiotics. Also, the purine riboswitches provide new opportunities for harnessing these RNAs for synthetic biological applications since these riboswitches can be reprogrammed to recognize compounds, such as guanine derivatives [26], pyrimidine [27], and pterin [28], with only a limited set of mutations. Finally, we expect our findings to contribute to the molecular basis information to better understand the dynamics of 2′-dG riboswitches and, possibly, as potential antibacterial targets.

## 4. Materials and Methods

### 4.1. Construction and Analysis of Molecular Systems

Atomic coordinates of the 2′-dG-I riboswitches complexed with 2′-deoxyguanosine (dG), and guanosine (rG) were obtained from the Protein Data Bank (PDB ID: 3SKI and 3SKZ [11]). The 2′-dG-II riboswitch bound to dG was retrieved from PDB ID: 6P2H [12]. The dG in the binding pocket of the 2′-dG-II riboswitch was replaced by rG to obtain the new complex. For all complexes, all crystallographic water molecules and Mg^2+^ cations were retained. Corresponding secondary structure information was taken from the PDB files through RNApdbee [29,30]. Sequences of both classes of 2′-dG riboswitches were then aligned using LocARNA [31], with default settings, and figures of the sequence alignment were rendered using ALINE [32]. Graphical representations of 2D and 3D structures were generated using VARNA [33] and Pymol [34], respectively.

### 4.2. Molecular Dynamics Simulations

Molecular dynamics (MD) simulations were carried out using AMBER 20.0 [35], and RNA interactions were represented using the ff99bsc0χOL3 force field [36,37]. Bonded, electrostatic and Lennard-Jones parameters for dG and rG ligands were obtained using the Generalized Amber force field (GAFF) [38] and AM1-BCC [39] tools, while atomic partial charges were calculated using ANTECHAMBER [40]. Electrostatic interactions were treated using the particle mesh Ewald (PME) algorithm with a cut-off of 12 Å. Each system was simulated under periodic boundary conditions in a triclinic box whose dimensions were automatically defined considering 12 Å from the outermost RNA atoms in all cartesian directions. The simulation box was filled with TIP3P water molecules [41]. We added one chloride and 34 magnesium counterions to neutralize the systems.

Subsequently, a two-step energy minimization procedure was performed: (i) 2000 steps (1000 steepest descent + 1000 conjugate-gradient) with all heavy atoms harmonically restrained with a force constant of 5 kcal mol^−1^ Å^−2^; (ii) 5000 steps (2500 steepest descent + 2500 conjugate-gradient) without position restraints. Next, initial atomic velocities were assigned using the Maxwell-Boltzmann distribution corresponding to a temperature of 20 K. The systems were gradually heated up to 300 K over one nanosecond using the Langevin thermostat. All heavy atoms were harmonically restrained during this stage with a force constant of 10 kcal mol^−1^ Å^−2^. All systems were subsequently equilibrated during nine successive 500-ps equilibration simulations where position restraints approached zero progressively. After this period, all the systems were simulated with no restraints at 300 K in the Gibbs ensemble with a 1-atm pressure using isotropic coupling. All chemical bonds containing hydrogen atoms were restricted using the SHAKE algorithm [42] and the time step was set to 2 fs. We simulated four independent MD runs of 500 ns for each aptamer complex, using different initial velocities, totaling 8 μs of production simulation time.

### 4.3. Trajectory Analysis

The root-mean-square deviation (RMSD) and root-mean-square fluctuations (RMSF) values were calculated separately for the whole RNA and its substructures, after fitting only heavy atoms to their respective parts, taking the crystallographic structure as a reference. These analyses were carried out using the GROMACS package tools [43]. In addition, dynamic secondary structure and sugar pucker of the ligands along the trajectories were assessed using the Barnaba [44].

Hydrogen bonds (H-bond) were calculated for each system, concatenating four independent MD simulations’ last 50-ns trajectory. H-bond formation was defined using a geometric criterion with CPPTRAJ [45] in Amber. We considered a hit when the distance between two polar heavy atoms, with at least one hydrogen atom attached, was less than 3.5 Å and using an H-donor angle higher than 120°.

The enthalpy of 2′-dG riboswitch complexes was calculated by extracting the uncorrelated 500 snapshots from each MD simulation’s last 50-ns trajectory, using the MM/GBSA (molecular mechanics generalized Born surface area) approach. All water molecules and counterions of each snapshot were stripped before the MM/GBSA calculation. The interaction energy and solvation free energy for the complex, receptor, ligand, and resulting averages were calculated using the MMPBSA.py module [46] available in the AMBER distribution. Finally, the conformational entropic contribution to the binding free energy was estimated for a total number of 50 snapshots using the normal mode analysis from each MD simulation’s last 50-ns trajectory.

The cross-correlation and network analyses were performed using the Bio3D and the *igraph* R packages [47]. The dynamic cross-correlation matrices (DCCM) [19,48,49] were initially calculated separately for each simulation using as inputs the corresponding MD trajectory superimposed onto the crystallographic structure. Then, each group of four matrices per riboswitch complex was utilized to obtain a consensus matrix. Finally, a proximity/contact map filter was applied to build the correlation network so that all systems had the same number of edges (938 edges). Briefly, graphs were obtained considering heavy atoms as nodes, and the connection between nodes *i* and *j* was weighted using the absolute values of cross-correlations (*C*_(*i,j*)_) coefficients (Equation (1)):(1)$$w_{\left(i,j\right)}=-log\left(\left|C_{\left(i,j\right)}\right|\right).\;$$

Yen’s algorithm [50] was used to calculate the shortest pathways connecting two nodes in the network. Path lengths are defined as the sum of the edge weights connecting a pair of nodes in each pathway. The first 1000 shortest paths were collected and employed to calculate the node degeneracy value, representing the percentage of pathways from the overall ensemble in which a given node is present.

## Figures and Tables

**Figure 1 ijms-23-01925-f001:**
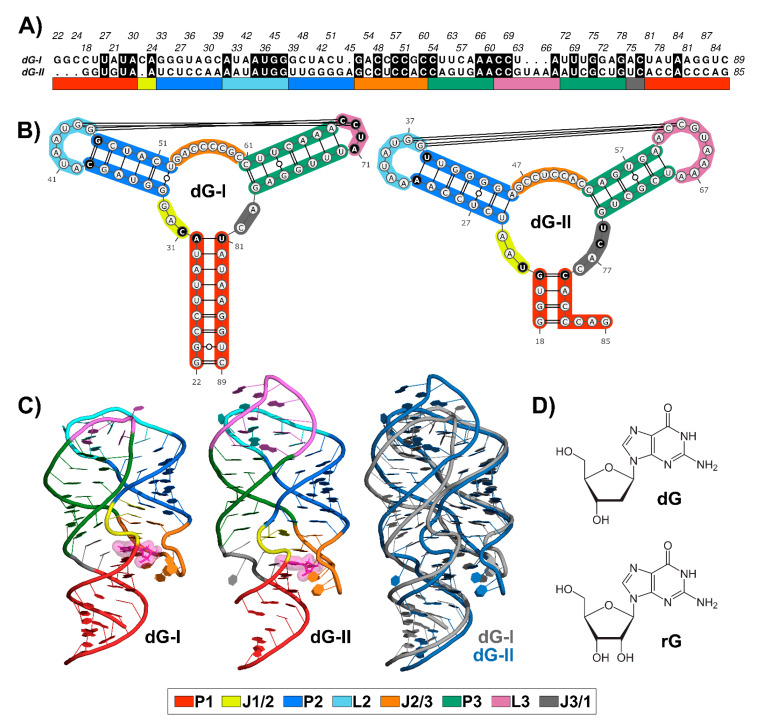
Sequence alignment and structures of 2′-dG-I and 2′-dG-II riboswitches classes. (**A**) Sequence alignment between 2′-dG riboswitches classes. Black filled positions represent conserved residues. (**B**) Secondary structure between 2′-dG riboswitches classes. Different key nucleotides between classes were colored in black. (**C**) Three-dimensional structure between 2′-dG riboswitches classes bound to cognate dG (van der Waals representation in pink). (**D**) Chemical structure of cognate dG and the noncognate rG. Stems, loops, and junctions were identified according to the figure legend caption.

**Figure 2 ijms-23-01925-f002:**
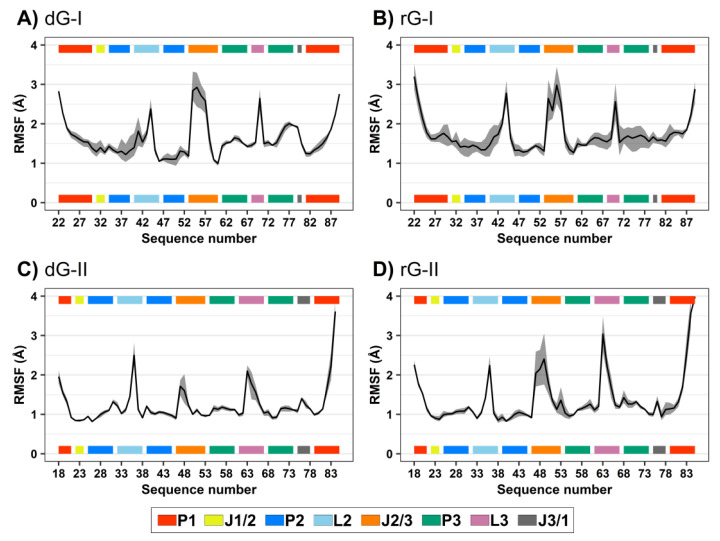
Root-mean-square fluctuations (RMSF) values of 2′-dG-I (**A**,**B**) and 2′-dG-II (**C**,**D**) riboswitches classes bound to dG (**A**,**C**) and rG (**B**,**D**). The aptamer substructures are depicted at the margin of the plots and colored according to the figure caption.

**Figure 3 ijms-23-01925-f003:**
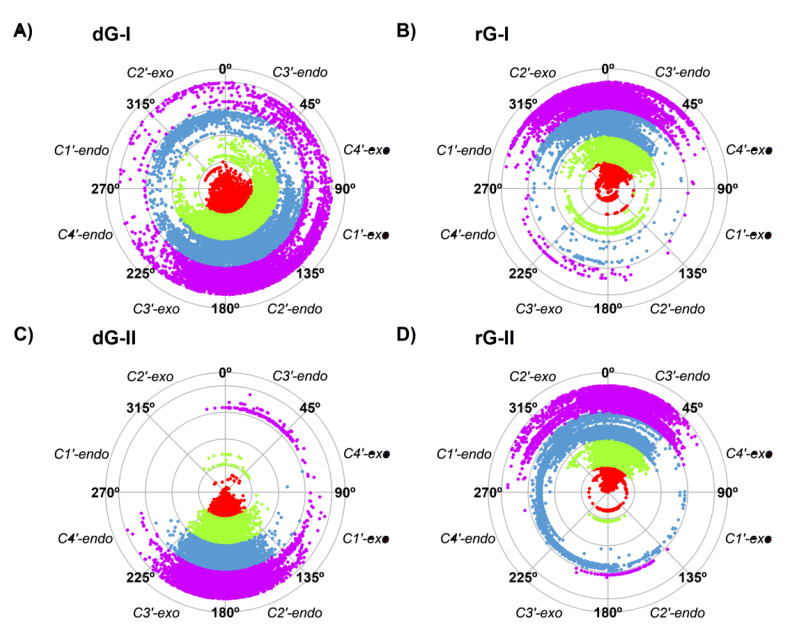
Sugar pucker of dG (**A**,**C**) and rG (**B**,**D**) ligands of 2′-dG-I (**A**,**B**) and 2′-dG-II (**C**,**D**) riboswitches systems. Each color in the plot represents a different replicate.

**Figure 4 ijms-23-01925-f004:**
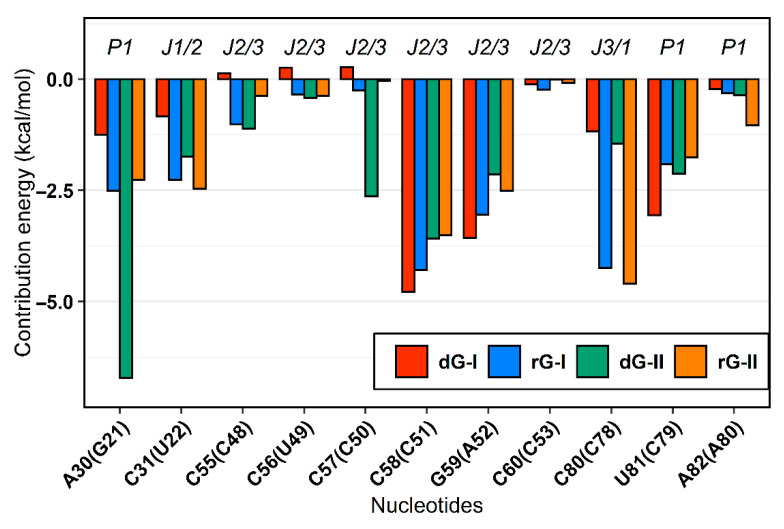
MM/GBSA per-nucleotide decomposition energies. Only nucleotides with energy less than zero were represented. The aptamer substructures are depicted at the top margin. Each system was colored according to the figure caption.

**Figure 5 ijms-23-01925-f005:**
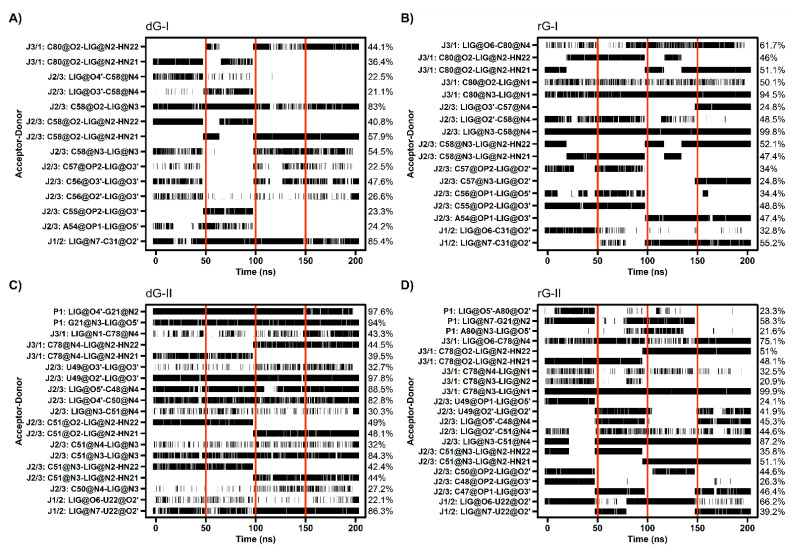
Hydrogen bond between 2′-dG-I (**A**,**B**) and 2′-dG-II (**C**,**D**) riboswitches with dG (**A**,**C**) and rG (**B**,**D**) ligands. The occupancy of each interaction is on the right axis. The calculation was performed for the last 50 ns of each replicate, which was separated by red vertical lines.

**Figure 6 ijms-23-01925-f006:**
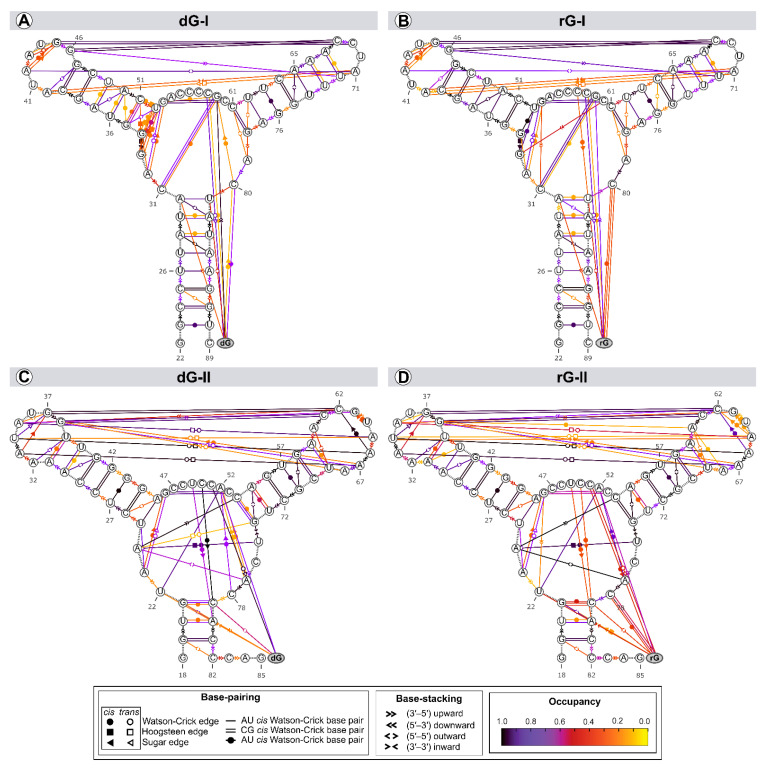
Dynamic secondary structure representation of 2′-dG-I (**A**,**B**) and 2′-dG-II (**C**,**D**) riboswitches classes bound to dG (**A**,**C**) and rG (**B**,**D**). The extended secondary structure annotation follows the Leontis-Westhof classification (see legend caption). The color scheme shows the fraction of frames (occupancy) within a system for which the interaction was formed.

**Figure 7 ijms-23-01925-f007:**
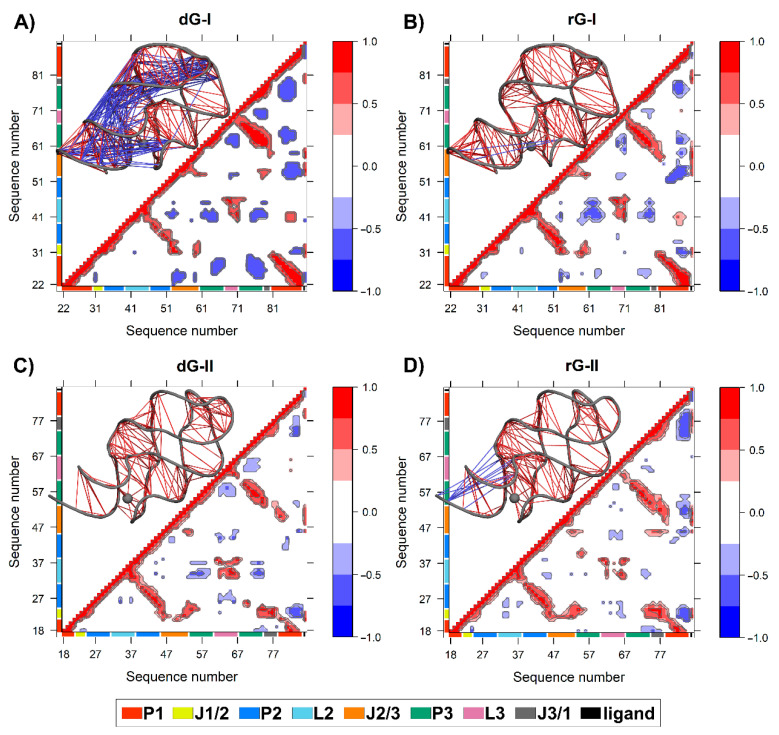
The dynamic cross-correlation matrices (DCCM) of 2′-dG-I (**A**,**B**) and 2′-dG-II (**C**,**D**) riboswitches classes bound to dG (**A**,**C**) and rG (**B**,**D**). Next to each matrix, the corresponding 3D structures with lines connecting pairs of correlated residues are shown. Only the pairs presenting (|Cij|) > 0.6. are represented for clarity’s sake. The aptamer substructures are depicted at the margin of the plots and colored according to the figure caption.

**Figure 8 ijms-23-01925-f008:**
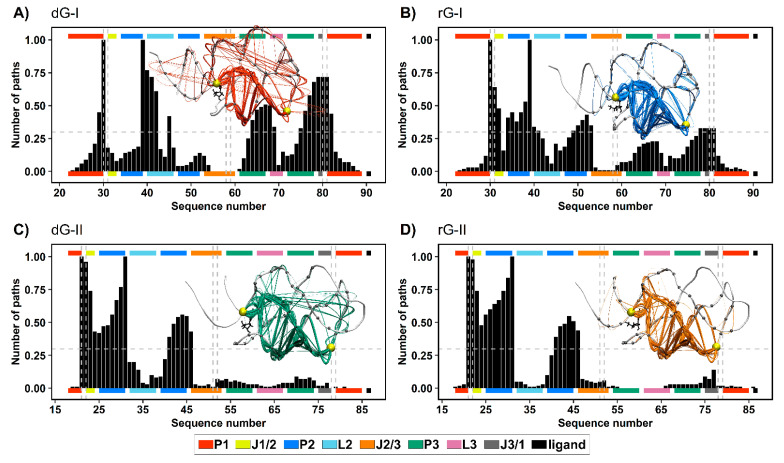
Normalized node degeneracy graph of 1000 shortest communication paths connecting A30(G21) and C39(A31) residues of 2′-dG-I (**A**,**B**) and 2′-dG-II (**C**,**D**) riboswitches classes bound to dG (**A**,**C**) and rG (**B**,**D**); and visualization of sub-optimal paths (with 200 shortest paths) in a correlation network. The aptamer substructures are depicted at the margin of the plots and colored according to the figure caption.

**Figure 9 ijms-23-01925-f009:**
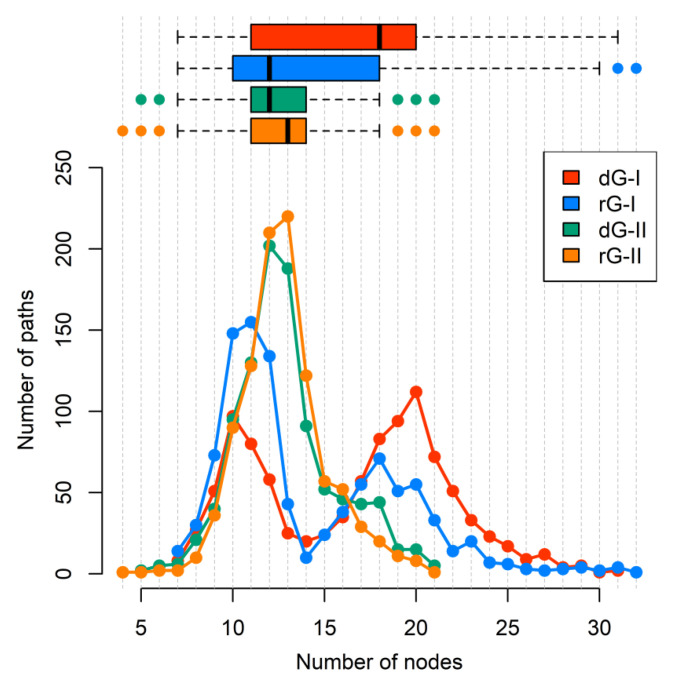
The number of nodes per path connecting A30(G21) and C39(A31) residues of 2′-dG-I and 2′-dG-II riboswitches classes bound to dG and rG. Each system was colored according to the figure caption.

**Table 1 ijms-23-01925-t001:** Root-mean-square deviations of 2′dG riboswitches as a whole and for substructures bound with dG and rG ^a^.

	2′dG-I		2′dG-II	
	dG	rG	dG	rG
RNA	3.09 ± 0.65	2.62 ± 0.61	2.98 ± 0.65	2.53 ± 0.40
P1	3.17 ± 0.75	2.43 ± 0.75	3.35 ± 1.11	1.30 ± 0.31
J1/2	4.39 ± 0.80	2.62 ± 0.70	1.57 ± 0.47	1.37 ± 0.32
P2	2.26 ± 0.45	2.49 ± 0.73	1.34 ± 0.25	1.30 ± 0.31
L2	2.53 ± 0.64	2.42 ± 0.62	1.86 ± 0.52	1.91 ± 0.41
J2/3	3.87 ± 0.94	3.38 ± 0.86	1.76 ± 0.46	2.77 ± 1.07
P3	2.73 ± 1.02	2.26 ± 0.73	1.93 ± 0.28	1.85 ± 0.26
L3	3.55 ± 0.77	3.19 ± 0.59	2.08 ± 0.39	2.95 ± 0.70
J3/1	2.96 ± 1.13	1.97 ± 0.66	2.30 ± 0.50	1.57 ± 0.35
Ligand	3.03 ± 1.00	2.12 ± 0.58	1.68 ± 1.00	2.23 ± 1.13

^a^ All values are given in Å. Standard errors of the average values were labeled by the ± signs.

**Table 2 ijms-23-01925-t002:** Binding free energies for complexes of 2′-dG riboswitches calculated by MM/GBSA method ^a^.

Items	2′dG-I		2′dG-II	
	dG	rG	dG	rG
ΔE_vdw_	−41.82 ± 0.08	−42.38 ± 0.10	−46.73 ± 0.07	−40.89 ± 0.09
ΔE_ele_	−22.52 ± 0.27	−61.31 ± 0.21	−30.00 ± 0.20	−76.67 ± 0.32
ΔG_egb_	38.54 ± 0.25	70.23 ± 0.19	40.08 ± 0.18	86.45 ± 0.29
ΔG_esurf_	−3.72 ± 0.004	−4.26 ± 0.003	−4.14 ± 0.004	−4.34 ± 0.004
^b^ ΔG_ele+egb_	16.01 ± 0.26	8.92 ± 0.20	10.08 ± 0.19	9.78 ± 0.30
^c^ ΔH	−29.53 ± 0.08	−37.72 ± 0.13	−40.79 ± 0.09	−35.44 ± 0.10
^d^ −TΔS	23.04 ± 0.44	21.64 ± 1.03	23.47 ± 0.85	23.54 ± 1.44
ΔG_bind_	−6.49	−16.08	−17.32	−11.90
^e^ ΔG_exp_	−10.97	−8.64	−11.62	−11.04

^a^ All mean and standard error values are given in kcal/mol. ^b^ ΔG_ele+egb_ = ΔE_ele_ + ΔG_egb_. ^c^ ΔH = ΔE_vdw_ + ΔE_ele_ + ΔG_esurf_ + ΔG_egb_. ^d^ T = 298.15 K. ^e^ The experimental values are from references [11,12].

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
