# Peer review of "Bacterial 2′-Deoxyguanosine Riboswitch Classes as Potential Targets for Antibiotics: A Structure and Dynamics Study"

_ijms, 2022, doi:10.3390/ijms23041925_

Round 1

Reviewer 1 Report

Two different 2’-17 deoxyguanosine (2′-dG) riboswitches, including Class I (2’-dG-I) and class II (2’-dG-II), are used as promising target of antibacterial drug design. The authors applied multiple simulation technologies to investigate binding mechanism of 2’-deoxyguanosine and riboguanosine toward 2’-dG-I and 2’-dG-II. Their results suggest that the stability and increased interactions in the three-way junction of 2’-dG riboswitches are responsible for a higher nucleoside ligand affinity. This work provides significant information for understanding dynamics changes of 2’-17 deoxyguanosine (2′-dG) riboswitches due to ligand bindings. I recommend that this work can be published in international journal of molecular science after a major revision.

  1. The salt strength highly affects conformational changes of riboswitches. Thus the authors should clarify the salt strength that they used in this work and point out the number of magnesium and chloride ions.
  2. GROMACS package tools and CPPTRAJ in Amber 20 were used to analyze MD trajectories. The authors should explain which information is analyzed by using GROMACS package tools and which information is analyzed by using CPPTRAJ.
  3. The authors used dynamics cross-correlation matrices to efficiently probe motion modes of riboswitches, but a work (DOI: 10.1021/acs.jcim.0c01470) on DCCM should be mentioned.
  4. The authors calculated RMSD and RMSFs of riboswitches and the authors should clarify which atoms are used to compute RMSD and RMSFs.
  5. In calculations of binding free energies, the authors should introduce how many snapshots are adopted to binding enthalpy and entropy. Moreover the authors clarify the effect of enthalpy and entropy on ligand bindings.
  6. The authors should how the structural differences of 2’-deoxyguanosine and riboguanosine affect ligand-nucleotide interactions.

Author Response

  1. The salt strength highly affects conformational changes of riboswitches. Thus the authors should clarify the salt strength that they used in this work and point out the number of magnesium and chloride ions.

We added the number of magnesium and chloride ions in the Materials and Methods section. Now it reads (page 17, line 503):

"We added one chloride and 34 magnesium counterions to neutralize the systems."

  1. GROMACS package tools and CPPTRAJ in Amber 20 were used to analyze MD trajectories. The authors should explain which information is analyzed by using GROMACS package tools and which information is analyzed by using CPPTRAJ.

We used Gromacs package for RMSD and RMSF analysis and CPPTRAJ for H-bond calculation. This information is in the Materials and Methods section (page 18, lines 521 to 529).

"The root-mean-square deviation (RMSD) and root-mean-square fluctuations (RMSF) values were calculated separately for the whole RNA and its substructures, after fitting only heavy atoms to their respective parts, taking the crystallographic structure as a reference. These analyses were carried out using the GROMACS package tools [43]"

"Hydrogen bonds (H-bond) were calculated for each system, concatenating four independent MD simulations' last 50 ns trajectory. H-bond formation was defined using a geometric criterion with CPPTRAJ [45] in Amber."

  1. The authors used dynamics cross-correlation matrices to efficiently probe motion modes of riboswitches, but a work (DOI: 10.1021/acs.jcim.0c01470) on DCCM should be mentioned.

Thanks for the reference. We have added it to the manuscript (page 18, line 542).

  1. The authors calculated RMSD and RMSFs of riboswitches and the authors should clarify which atoms are used to compute RMSD and RMSFs.

We use heavy atoms for RMSD and RMSF calculations (any non-hydrogen atom). This information can be found in the Results (page 4, line 137) and Methodology (page 18, line 548) sections.

"We regarded only the heavy atoms for calculations over the trajectory production stage, taking as references the crystallographic structures (Table 1 and Figure 2)."

"The root-mean-square deviation (RMSD) and root-mean-square fluctuations (RMSF) values were calculated separately for the whole RNA and its substructures, after fitting only heavy atoms to their respective parts, taking the crystallographic structure as a reference."

  1. In calculations of binding free energies, the authors should introduce how many snapshots are adopted to binding enthalpy and entropy. Moreover the authors clarify the effect of enthalpy and entropy on ligand bindings.

We used 500 and 50 snapshots per MD simulation to calculate enthalpy and entropy, respectively (see Materials and Methods section, page 18, lines 531 to 539). In our simulations, the -TΔS term resulted in comparable values for all systems, so the divergences in ΔGbind observed among the systems were due to the enthalpic term, and consequently, the binding process was enthalpically driven (page 7, lines 197 to 200).

"The enthalpy of 2'-dG riboswitches complexes was calculated by extracting the uncorrelated 500 snapshots from each MD simulation last 50 ns trajectory, using the MM/GBSA (molecular mechanics generalized Born surface area) approach. All water molecules and counterions of each snapshot were stripped before the MM/GBSA calculation. The interaction energy and solvation free energy for the complex, receptor, ligand, and resulting averages were calculated using the MMPBSA.py module [46] available in the AMBER distribution. Finally, the conformational entropic contribution to the binding free energy was estimated for a total number of 50 snapshots using the normal mode analysis from each MD simulation's last 50 ns trajectory."

"In our simulations, the entropic contributions (−T∆S) resulted in comparable values for both systems (~23 kcal/mol), so the divergences in ΔGbind observed among the classes were due to the enthalpic term, and consequently, the binding process was enthalpically driven."

  1. The authors should how the structural differences of 2’-deoxyguanosine and riboguanosine affect ligand-nucleotide interactions.

It was impossible to verify, regarding only the ligand structures, how stereochemically differences between ligands affect the interactions with the receptor because the obtained data are not conclusive. However, we observed that the difference is about ~5 kcal/mol in interaction energy. This could be due to the loss of a polar interaction regarding the chemical environment context. Nonetheless, we tried to explore as much as possible each system's particularities and make a counterpoint to the work already published (Discussion section).

Reviewer 2 Report

This is a well prepared and interesting study. It is acceptable in this form, but two points can be changed in the manuscript. 

Comments:

 Page 2, line 87-92: This part is a conclusion. At the end of introduction part only the aim of the study should be indicated.

In the title of Table 2, 2′ dg should be changed to dG

Author Response

Page 2, line 87-92: This part is a conclusion. At the end of introduction part only the aim of the study should be indicated.

We appreciate your comments, but we have formatted the manuscript following the journal guidelines, which suggest that the introduction section highlights the principal conclusions of the work. Below are the guidelines stated by the journal.

"The introduction should briefly place the study in a broad context and highlight why it is important. It should define the purpose of the work and its significance. The current state of the research field should be carefully reviewed and key publications cited. Please highlight controversial and diverging hypotheses when necessary. Finally, briefly mention the main aim of the work and highlight the principal conclusions. As far as possible, please keep the introduction comprehensible to scientists outside your particular field of research. References should be numbered in order of appearance and indicated by a numeral or numerals in square brackets—e.g., [1] or [2,3], or [4–6]. See the end of the document for further details on references."

In the title of Table 2, 2′ dg should be changed to dG

We replaced "2'dg riboswitches" instead of "2'-dG riboswitches" (page 7, line 204).

Reviewer 3 Report

The article is interesting and helpful for research in wet labs. It is pleasure to read the study.

Author Response

We appreciate the reviewer's comment.

Round 2

Reviewer 1 Report

The authors have responsed all my issues, thus I recommend that this work can accepted for publishing at International journal of molecular science.